# Optimized single EEG channel selection for detecting major depressive disorder

## Abstract

Major depressive disorder (MDD) or depression is a chronic mental illness that significantly impacts individuals' well-being and is often diagnosed at advanced stages, increasing the risk of suicide. Current diagnostic practices, which rely heavily on subjective assessments and patient self-reports, are often hindered by challenges such as under-reporting and the failure to detect early, subtle symptoms. Early detection of MDD is crucial and requires monitoring vital signs in everyday living conditions. Electroencephalogram (EEG) is a valuable tool for monitoring brain activity, offering critical insights into MDD and its underlying neurological mechanisms. While traditional EEG systems typically involve multiple channels for recording, making them impractical for home-based monitoring, wearable sensors can effectively capture single-channel EEG data. However, generating meaningful features from this data poses challenges due to the need for specialized domain knowledge and significant computational power, which can hinder real-time processing. To address these issues, our study focuses on developing a deep learning model for the binary classification of MDD using single-channel EEG data. We focused on specific channels from various brain regions, including central (C3), frontal (Fp1), occipital (O1), temporal (T4), and parietal (P3). Our study found that the channels Fp1, C3, and O1 achieved an impressive accuracy of 88% when analyzed using a Convolutional Neural Network (CNN) with leave-one-subject-out cross-validation. Our study highlights the potential of utilizing single-channel EEG data for reliable MDD diagnosis, providing a less intrusive and more convenient wearable solution for mental health assessment.

## 1 Introduction

Major depressive disorder (MDD), a widespread condition affecting individuals globally is characterized by persistent feelings of sadness, hopelessness, and a lack of interest or pleasure in daily activities [Marx et al. (2023)], often accompanied by suicidal thoughts or attempts [American Psychiatric Association et al. (2013); Tiller (2013); Australian Bureau of Statistics (2007); Obuobi-Donkor et al. (2021)]. MDD is particularly prevalent among individuals aged 18-29 and affects an estimated 350 million people worldwide [Organization et al. (2008)]. Notably, there have been reported increases in MDD cases during the COVID-19 pandemic [Lawrence et al. (2015); Lakhan et al. (2020)]. This highlights the urgent need for effective screening and intervention strategies in this age group. To enable timely intervention and effective treatment, there is a pressing need for accurate and objective diagnostic tools for MDD. This could be achieved through the monitoring of specific physiological signals in real-world settings using wearable devices.

Diagnosing MDD primarily relies on clinician-administered and self-rated scales, which are subjective and unsuitable for continuous monitoring [American Psychiatric Association et al. (2013)]. Experience-based diagnoses can be inaccurate due to variability in individual experiences. MDD affects various regions of the brain [Pandya et al. (2012)], with key areas such as the dorsal and medial prefrontal cortex, dorsal and ventral anterior cingulate cortex, orbital frontal cortex, hippocampus, insula, and amygdala playing crucial roles [Drevets (2000; 1999a); Buchsbaum et al. (1997)]. Electroencephalogram (EEG) which measures the brain's electrical activity [Carney et al. (2001)] stands out as a superior biomarker for MDD detection as it directly measures brain activity and connectivity, offering insights into mood regulation and cognition [Olbrich & Arns (2013); Mumtaz et al. (2015)]. Using more electrodes in EEG recordings enhances coverage of brain activ-

ity across different regions, but it also increases complexity and can reduce patient comfort due to the need for conductive gel and hair preparation [Montoya-Martínez et al. (2021)]. While clinical settings can handle the use of multiple electrodes, wearable devices prioritize simplicity, often using fewer channels, which can compromise data quality and accuracy. The challenge lies in selecting the most effective channels, requiring thorough research to ensure fast and reliable data collection [Alotaiby et al. (2015)]. A promising solution is the development of machine/deep learning models that can accurately diagnose the medical condition using data from fewer electrodes, making the process more efficient and practical.

Some studies have investigated the use of deep learning in MDD diagnosis using EEG data, but the channel reduction remains less explored. For example, Acharya et al. (2018) employed a 13-layer Convolutional Neural Network (CNN), while Ay et al. (2019) combined CNN with Long Short-Term Memory (LSTM) models to capture both spatial and temporal patterns in EEG data. Using ten-fold cross-validation, these models demonstrated impressive accuracy, with Acharya et al. (2018) achieving 96% accuracy and Ay et al. (2019) reaching 99.12% for right hemisphere EEG data. The performance improvement in Ay et al. (2019) could be attributed to the inclusion of LSTM, which is designed for sequential learning and is well-suited for handling the temporal dynamics inherent in EEG signals. Contrary to this trend, Mumtaz & Qayyum (2019) compared the performance of a 11-layer 1D CNN and a 14-layer 1D CNN-LSTM model on the same dataset, achieving 98.32% and 95.97% accuracy, respectively, using ten-fold cross-validation. Interestingly, their results showed a higher accuracy for the CNN-only model. Khan et al. (2021) used partial directed coherence (PDC) to estimate effective brain connectivity for MDD classification. They trained a 3D CNN with PDC matrices, achieving 100% accuracy using 10-fold cross-validation. In a later study Khan et al. (2022) applied wavelet coherence (WCOH) on EEG segments, achieving 100% accuracy with a 2D-CNN model, but noted that further assessments are needed to validate these findings. Saeedi et al. (2021) introduced five deep learning models for MDD detection, focusing on brain connectivity analysis. Their 1DCNN-LSTM model outperformed others, achieving 99.25% accuracy. Dang et al. (2020) developed a frequency-dependent multi-layer brain (FDMB) network paired with a CNN model to classify MDD from EEG signals and achieved an accuracy of 97.27%. Loh et al. (2022) suggested a CNN model for MDD detection using spectrogram images from EEG data. These images were passed through an eight-layer CNN network, achieving 99.58% accuracy with hold-out validation. However, they acknowledged that 2D-CNN models can be computationally expensive, making clinical implementation challenging. The only deep learning study where it talked about channel reduction is by Rafiei et al. (2022), but again the reduced number of channels is 10, which is again a multichannel EEG. The study introduces a customized InceptionTime model for automated detection of MDD using raw EEG signals, achieving 91.67% accuracy with 19 channels and 87.5% with 10 channels and with the first minute of EEG recordings.

The only single channel detection found is in the classical machine learning technique where there is a requirement of feature domain knowledge. For example, Bachmann et al. (2018) identified the EEG channel Pz as the most relevant, achieving 92% accuracy with 13 subjects in each class. Deep learning, unlike classical machine learning, excels at automatically learning complex representations from raw data without the need for manual feature engineering [LeCun et al. (2015)]. In classical machine learning, feature engineering is often labor-intensive and requires domain expertise, which can limit model performance by relying on predefined features [Schmidhuber (2015)]. Deep learning's ability to discover intricate patterns directly from data has led to breakthroughs in fields such as medical diagnostics, where it outperforms traditional methods by capturing subtle, high-dimensional relationships [Goodfellow (2016)]. The aim of our study is to investigate the applicability of single channel EEG data for MDD classification using deep learning.

The primary contribution of this study lies in the detection of MDD using a single EEG channel by leveraging CNN model. Unlike traditional approaches that rely on multiple channels, this study demonstrates the feasibility of achieving accurate diagnosis with minimal electrode usage, which is crucial for wearable applications. Our code is available online in the GitHub (link will be shared after review).

## 2 METHODOLOGY

### 2.1 PROBLEM FORMULATION

The classification of MDD from EEG signals is handled as a binary classification problem, where a CNN model is trained to distinguish between MDD (labeled as 1) and non-MDD (labeled as 0) based on EEG inputs of a specified duration. Once the model is trained, it produces binary predictions for each input signal. An EEG recording is classified as MDD if the proportion of segments predicting MDD surpasses a threshold of x% compared to the total segment predictions for that subject. As depicted in Figure 1, the process starts with EEG signal acquisition, followed by pre-processing to remove artifacts and segment the data. The deep learning model is then used to classify the signals as originating from either MDD or non-MDD individuals.

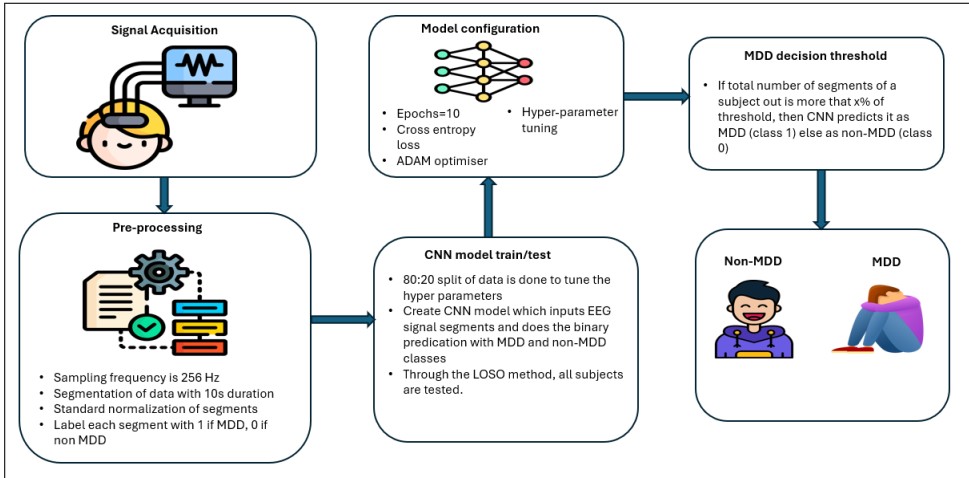

Figure 1: Flow diagram of MDD classification from single channel EEG signal using CNN model.

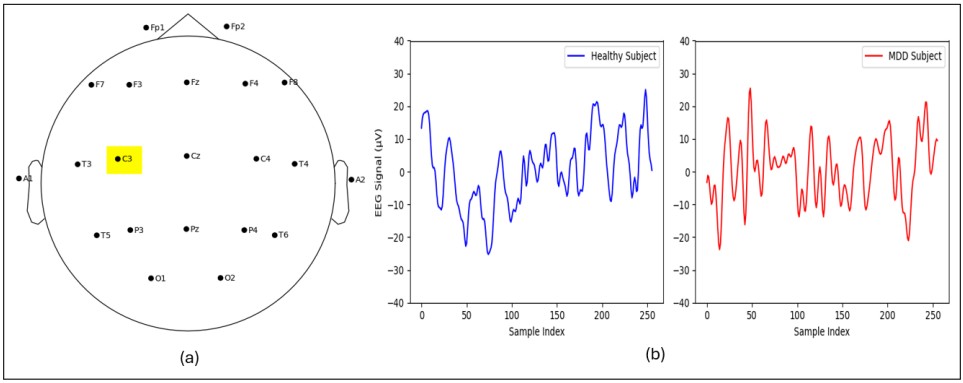

Figure 2: Electrode placement and EEG signal comparison between subjects with MDD and non-MDD. (a) Electrode placement for EEG recording used in this study, highlighting the central region channel C3. (b) EEG signal recorded at the C3 channel over one second, displaying sample indices from 0 to 255 at a sampling rate of 256 Hz. Differences in signal patterns are observed between MDD and non-MDD subjects.

### 2.2 DATA

The dataset used in this study was obtained from Mumtaz et al. (2017), and is publicly accessible at the following open-access URL: https://figshare.com/articles/dataset/EEG_Data_New/4244171. The study protocol received ethical approval from the Ethics Committee at

Hospital University Sains Malaysia (HUSM). The EEG recordings included 30 patients diagnosed with MDD, aged 27 to 53 years (mean age = 40.3 ± 12.9), and 28 control subjects, aged 22 to 53 years (mean age = 38.3 ± 15.6). All participants were recruited from the outpatient clinic of HUSM, with informed consent obtained prior to participation. Participants are selected based on the Diagnostic and Statistical Manual of Mental Disorders, Fourth Edition (DSM-IV) criteria. Exclusion criteria included pregnancy, alcoholism, smoking, and epilepsy. The EEG data were recorded at a sampling rate of 256 Hz for 5 minutes while participants were in a resting state with their eyes closed. The recordings spanned 19 channels, covering central, frontal, parietal, occipital, and temporal regions. The Figure 2 shows the electrode placement for EEG recording used in this study and the variation of EEG signal amplitude of MDD and non-MDD subjects from a single channel. Even though we have used 5-minute signal in our entire study, the plot is only for one second duration from channel C3.

## 2.3 SEGMENTATION

In our study, the 5-minute EEG recordings were segmented into non-overlapping 10-second segments. Using shorter segments, such as 10 seconds, is effective for capturing the temporal variability and transient features associated with MDD [Li et al. (2016)]. Although some subjects' recordings slightly exceed the 5-minute duration, all data were included in the analysis.

## 2.4 NORMALIZATION

For each channel, the extracted features were normalized using the standard scaler method. This method transforms the features to have a mean of 0 and a standard deviation of 1, ensuring that all features are on a uniform scale and reducing the impact of outliers. The standard scaler is computed using the formula:

$$z = (x - m)/s \tag{1}$$

where as x is the feature, m is the mean of the feature and s is the standard deviation.

## 2.5 HYPER-PARAMETER TUNING

In this study, we used a deep learning model for binary classification using a sequential architecture designed to analyze time-series data. In our study, we employ hyper-parameter tuning using the 80:20 training/validation split of the total segments. Hyper-parameters were optimized for:

- kernel size among the lengths 3, 5, 7, 11, 21 and 31,
- number of max pooling layers (number of convolution block) with blocks 2, 3, 4 and 5, and
- MDD decision threshold across range 10-100% with 10% step.

To classify an EEG recording as indicative of MDD, we defined a prediction threshold, denoted as x% for the MDD-predicting segments. If the number of segments classified as MDD (class 1 by the deep learning model) exceeds x% of the total predictions, the EEG recording is labeled as MDD; otherwise, it is classified as non-MDD. The threshold x% can range from 10% to 90%, allowing for adjustments in the model's sensitivity. A lower threshold (e.g., 10%) increases the likelihood of classifying subjects as MDD, while a higher threshold (e.g., 90%) biases the classification towards healthy subjects. The decision threshold hyper-parameter was searched for 80:20 subject-wise training/validation split. The threshold that yielded the highest accuracy was identified through multiple validation iterations, and this final value was used for leave-one-subject-out (LOSO) testing to evaluate performance.

## 2.6 CONVOLUTIONAL NEURAL NETWORK (CNN) MODEL

The CNN model used for single-channel EEG based MDD classification is as shown in Figure 3. The model begins with an input layer that accepts single-channel EEG data. The model consists of three convolutional layers (Conv1D) with progressively increasing filter sizes of 64, 128, and 256, each utilizing a kernel size of 21 and a ReLU activation function to capture intricate features from the input signals. After each convolutional layer, a MaxPooling1D layer with a pool size of 2 is applied to reduce the dimensionality of the feature maps, enhancing computational efficiency

while preserving essential information. Following the convolutional and pooling layers, the output is flattened to prepare for the final dense layer. The model culminates in a single dense output layer with a sigmoid activation function, which is designed for binary classification.

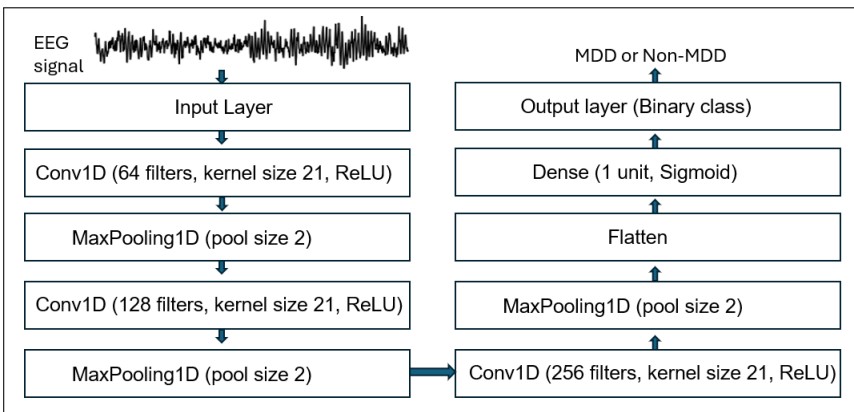

Figure 3: The CNN architecture for MDD classification from single channel EEG signal.

### 2.7 Model Training and Testing

In our study, we employed a leave-one-subject-out cross-validation approach using a CNN model. In LOSO cross-validation, $N-1$ subjects are utilized for training, while one subject is set aside for testing. This method ensures that each subject's data is used exclusively for testing exactly once, thus preventing any data leakage between the training and testing phases. By adopting LOSO cross-validation, we ensured a more effective model training process and a robust evaluation of the model's performance.

### 2.8 Metric used

We utilize accuracy as the primary evaluation metric for our deep learning model due to the balanced nature of our dataset, where the classes (e.g., presence and absence of MDD) are approximately equal in size. In balanced datasets, accuracy provides a reliable measure of model performance because each class contributes equally to the calculation, ensuring the metric is not skewed by class predominance. Accuracy is straightforward to interpret, offering a clear assessment of overall effectiveness, which is accessible to clinicians and researchers. High accuracy indicates reliable identification of both conditions, crucial for clinical decision-making and patient care, making it a robust and practical choice for evaluating our models in MDD detection using EEG signals.

## 3 Results

Figure 4 presents the variation of classification accuracy using the CNN model with channel C3 EEG data and different kernel sizes and number of max pooling layers. From the Figure 4 (a), we can observe that a kernel size of 21 is the optimum for the CNN model. This suggests that the chosen kernel size is well-suited for capturing relevant features in the EEG data.

By selecting the optimum kernel size, the next step was carried out to see the impact of number of max pooling layers. For that we have checked the variation of accuracies among different number of max pooling layers such as 2, 3, 4 and 5. We could see that 3 servers as the optimum number and the result is as shown in Figure 4 (b).

The threshold is a crucial parameter that requires careful consideration. A lower threshold tends to bias predictions toward the healthy class, while a higher threshold skews results toward the MDD class. To minimize the false positive rate, a higher prediction threshold is generally preferred. In this study, the optimal threshold was determined to be 0.6, consistent with the threshold identified during the model's hyper-parameter tuning and validation process.

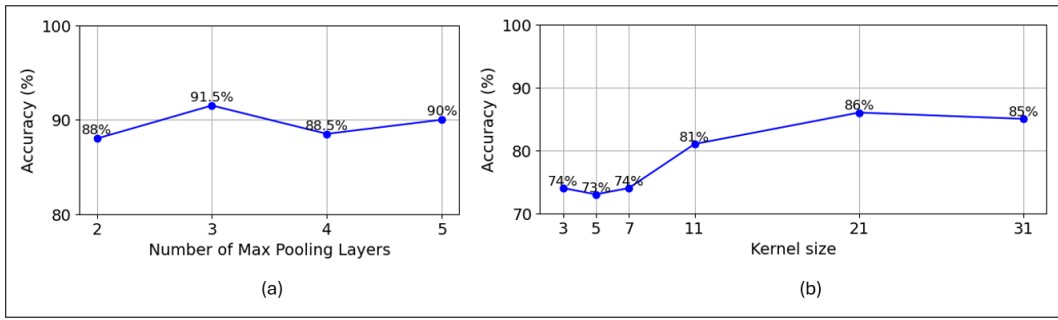

Figure 4: The variation of the classification accuracies according to the different kernel sizes and number of max pooling layers using the CNN model. Figure (a) is the accuracy variation with kernel sizes and Figure (b) is with the variation of accuracy with number of pooling layers.

The performance of the single -channel EEG based MDD detection using the proposed CNN model is evaluated. We used the epoch in CNN model equal to 10 and the segment duration as 10 s. We have selected single-EEG channel from each of the brain region such as frontal, central, temporal, occipital and parietal. The channels selected are Fp1, C3, T4, O1 and P3 from respective regions as mentioned above. Channels C3, O1 and Fp1 has shown the highest performance , which is 88%. The individual channel performance using the proposed CNN model is as shown in the Figure 5.

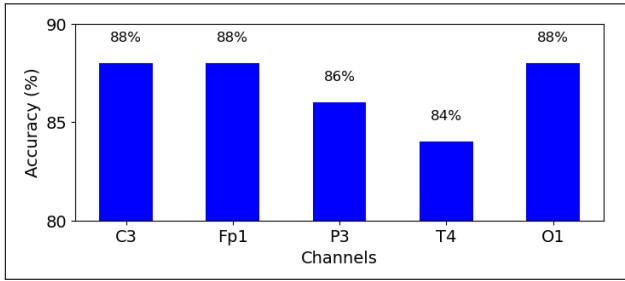

Figure 5: Performance of CNN model for MDD classification from single channel EEG signal from various brain regions.

## 4    DISCUSSION AND CONCLUSION

The primary finding of this study, based on the deep learning for classifying the MDD and non MDD include: 1) MDD and non MDD subjects can be classified from single channel EEG signals using deep learning models, 2) the single channel classification accuracy shows difference among the brain regions such as central, frontal, parietal, occipital and temporal, 3) the highest accuracy of classification is achieved in the frontal, central and occipital with 88% classification accuracy.

Although EEG has been widely studied in MDD research, the potential to distinguish MDD from non-MDD subjects using single-channel EEG remains under explored, particularly in the context of early diagnosis via wearable technology. While increasing the number of electrodes enhances the granularity of brain activity data, it also introduces challenges in terms of practicality and patient comfort, especially in clinical environments [Montoya-Martínez et al. (2021)]. Wearable devices, which prioritize minimal channels, underscore the need for research aimed at optimizing channel selection for MDD detection. Deep learning eliminates the need for expert-driven feature extraction, as it can automatically learn relevant patterns directly from raw EEG data. It has also demonstrated that sufficient information retrieval is achievable with fewer EEG channels, as seen in applications like seizure detection [Ullah et al. (2018)] and MDD detection [Acharya et al. (2018); Ay et al. (2019; Mumtaz & Qayyum (2019); Khan et al. (2022)]. While more electrodes in EEG recordings can provide deeper insights into brain activity, this approach can complicate the process and decrease patient comfort, especially in clinical settings [Montoya-Martínez et al. (2021)]. Wearable devices,

which prioritize fewer channels for ease of use, demand further research to identify the optimal channels for effective MDD screening.

MDD manifests in various brain regions [Drevets (1999b)], facilitating its differentiation through EEG variations between affected and unaffected individuals. While previous deep learning based EEG studies for MDD detection predominantly employ multichannel detection, ranging from 10 [Rafiei et al. (2022)] to 19 channels [Khan et al. (2021; 2022); Dang et al. (2020); Saeedi et al. (2021); Aydemir et al. (2021); Loh et al. (2022)]. Despite the extensive use of multiple channels spanning different brain regions, there is a lack of comparative analysis regarding region-wise performance. In our study, we have used single channel performance from each of the brain regions such as frontal, temporal, occipital, parietal, and central. We observed variations among channels across five distinct regions. Interestingly, selected EEG channels in our study exhibited high classification accuracy, suggesting their potential utility.

Rafiei et al. (2022) have investigated the impact of reducing EEG channels on MDD diagnosis using deep learning and the same dataset that we use in our study. However, the study uses a customized InceptionTime model for automated detection of MDD using 10 channel EEG signals and achieves an accuracy of 87.5% with first minute of EEG recordings. Even though the dataset is 5 minute duration they have used only the first one minute for this analysis. Using single channel EEG, our study yielded 88% accuracy in channel C3, Fp1 and O1 (shown in Figure 5).

The kernel size in a CNN model plays a key role in determining how much of the input data the model processes at a time [Alzubaidi et al. (2021)]. A smaller kernel size captures fine-grained, local features, which is useful for identifying detailed patterns in the data. In contrast, a larger kernel size can capture more global patterns by considering a broader context of the input. The choice of kernel size directly impacts the model's ability to recognize features at different scales [Agrawal & Mittal (2020); Phan et al. (2021)], and finding the optimal size is crucial for improving performance, especially in tasks like EEG signal classification where both local and global brain activity patterns may be important. The impacts of kernel size in classification accuracy is as shown in Figure 4 and the optimal kernel size is set as 21. The pooling layer in a CNN performs sub-sampling of feature maps generated by convolution operations, reducing their size while retaining key features [Alzubaidi et al. (2021)] and 3 max pooling layer was found optimum. In our analysis, three convolution blocks (a block consists of convolution and max pooling layers) were found suitable which yields a shallow CNN model (number of parameters $9,39,265$) which favors deploying in resource constrained devices.

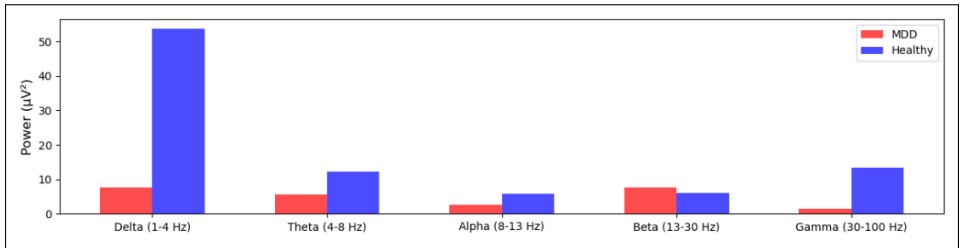

Figure 6: Band power variation among the MDD and non-MDD subjects in different EEG bands with signal duration of 10s.

It is important to see if there is distinction between the classes (MDD and non-MDD) when we use the segment duration as 10s. One way to look into is probably investigate the frequency distribution of EEG signals. When we analyse the EEG band wise (alpha, beta, gamma, delta and theta) power of the segments in non-MDD and MDD subjects, we can see the difference among the two classes such as all the bands show high power in healthy segments showing high variance in delta band, with more than $40 \mu V^2$, except the beta band which shows prominence of MDD segments (shown in the Figure 6). Certain frequency bands are associated with specific cognitive and emotional processes [Zheng & Lu (2015)]. For instance, alpha waves are linked to relaxation and inhibition, while beta waves are associated with alertness and active thinking [Roohi-Azizi et al. (2017)]. Changes in these frequencies can provide insights into how MDD affects cognitive and emotional processing [Hinrikus et al. (2010); Roh et al. (2016); Sun et al. (2008)]. The CNN model's performance can be

attributed with such discriminating features based on which the model is making the classification decision, however, this requires further investigation.

The study demonstrates that EEG channels from various brain lobes can effectively distinguish between normal and MDD subjects, highlighting the significance of electrode placement in areas that do not require hair removal. This finding is particularly relevant in today's fast-paced environment, where mental health concerns are often overlooked. Many individuals may be reluctant to undergo hair removal from the head, especially if they are unaware of or skeptical about their mental health status. Our study offers a non-invasive method for diagnosing MDD, providing a convenient and accessible way for individuals to assess their mental well-being. Notably, our model achieved an accuracy of 88% for the frontal channel Fp1, which is located near the forehead, making it more suitable for wearable applications. Further research is needed to validate these results across diverse MDD datasets and other channels.

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
