# OpenReview forum: "OPTIMIZED SINGLE EEG CHANNEL SELECTION FOR DETECTING MAJOR DEPRESSIVE DISORDER"
_ICLR.cc/2025/Conference — Submitted to ICLR 2025_

### Official Review · Reviewer_SFKv · 2024-10-16

**Soundness:** 1
**Presentation:** 1
**Contribution:** 1
**Rating:** 1
**Confidence:** 4

**Summary:**

This paper argues that Major Depression Disorder (MDD) should be diagnosed using a single EEG channel instead of the full electrode channels. However, I'm seriously concerned about the lack of contribution to this work. This conference is neither a medical nor a domain-specific journal for Major Depressive Disorder. I strongly recommend referring to the BCI paper presented at last year's ICLR (Jiang et al.), which offers a more relevant perspective.

[1] Large brain model for learning generic representations with tremendous EEG data in BCI, ICLR 2024

**Strengths:**

None.

**Weaknesses:**

1. None of the techniques proposed for single-channel selection:
The paper emphasizes the importance of single-channel selection from the title itself, yet it does not propose any method for selecting the channels. Instead, the authors used predefined channels and measured performance based on those. It is surprising that the paper does not consider commonly used methods for channel selection such as Hyperparameter Optimization (HPO), Layer-wise Relevance Propagation (LRP), or attention maps.
2. Lack of novelty of the model:
The paper uses a very generic CNN structure (3 layers of CNNs) without any optimization techniques specifically targeting single channels.
3. Weak Results section:
There are no competing methods presented in the results. Where are the competing methods mentioned in the Introduction?
4. Meaningless performance listing in the Introduction:
In the Introduction, the authors list various competing methods and their accuracies, but this serves no meaningful purpose. Simply listing accuracies from different papers applied to different datasets is not insightful.
5. Limited usage of the dataset (58 samples)

**Questions:**

1. If the single-channel-based classification is your contribution, why limit the study to MDD data? There are numerous datasets that use EEG. Why only use a dataset for MDD, which has a limited number of subjects?
2. In Figure 2(a), the electrode appears to be positioned outside the human head. What is this about?
3. The hyperparameter tuning was entirely empirical. Does this really need to be included in the Method section of the main text?
4. There are some prior studies that claim 100% or near-perfect accuracy in 10-fold cross-validation in your paper which raises the question, if such accuracy is achievable, why does this research even need to be conducted?

---

### Official Review · Reviewer_pAPb · 2024-10-28

**Soundness:** 1
**Presentation:** 2
**Contribution:** 1
**Rating:** 1
**Confidence:** 5

**Summary:**

The submitted manuscript explains the approach of Major Depressive Disorder detection with effective single channel EEG signal. Authors made an attempt to pick the efficient and potential channel for the mental health monitoring.

**Strengths:**

Addressing the potentiality of using single channel EEG in real-time implementation.

**Weaknesses:**

1. Lack of novelty!! In specific authors doesn't propose application dedicated features or tailored network configuration. Simply utilised the deep learning models to classify.
2. Recent publications proved that the understanding the underlying connectivity among the channels can give the gain in depression prediction efficacy. For instance ("Delaunay Triangulated Simplicial Complex Generation for EEG Signal Classification", "GM-VRC: Semantic Topological Data Ensemble Approach for EEG Signal Classification", "Chromatic Alpha Complex Generation for EEG Signal Classification").
3. Due to the high non-stationarity of the EEG signal behaviour, single channel utilization in neurological/ mental disorder identification is an uncertain task.
4. Presented manuscript lacks the presentation skills of the results obtained.
5. State of arts comparison with the irrelevant or out-dated articles, can be updated with the latest and advanced techniques of depression detection.
6. Can improve the presentation quality of the manuscript.

**Questions:**

1.	Main concern of the manuscript is channel selection. Authors selected few channels randomly; potential channels can be there in the channels left. At least, existing channel selection ranking scheme can be used for the same.
2.	Lack of novelty in the methodology utilized. Framework or methodology used in this manuscript follows the simple traditional implementation flow by utilizing existing well-established tools.
3.	No usage of the potential biomarker of single channel (temporal, spatial, frequency, statistical, and connectivity based). Using the potential features or biomarkers can improve the prediction accuracy.
4.	Usage of the existing simple CNN1D model!! At least implementation of the tailored network can be considered as novel contribution.
5.	Figure 2 shows the C3 electrode neural activity for 1 sec. Authors can discuss the time frame of that signal. Both are at the same interval or different intervals.
6.	Comparison of proposed method performance with the existing works can increase the quality of the manuscript.
7.	Authors claimed that Fp1, C3, and O1 can be the potential channels for the single channel detection of the depression classification. Did the validation of the same is done on another EEG dataset or not?
8.	Authors can also take care of listing the references, few references are repeated.

---

### Official Review · Reviewer_7RfZ · 2024-10-28

**Soundness:** 2
**Presentation:** 2
**Contribution:** 2
**Rating:** 3
**Confidence:** 4

**Summary:**

The paper Optimized Single EEG Channel Selection for Detecting Major Depressive Disorder proposes a CNN-based approach for diagnosing MDD using single-channel EEG data, aiming to support wearable technology for mental health monitoring. The authors test several EEG channels from different brain regions and find that specific channels, particularly from the frontal, central, and occipital regions, yield high classification accuracy (up to 88%) in differentiating MDD from non-MDD subjects. This approach prioritizes minimal electrode usage, aiming to make MDD monitoring feasible for home settings.

**Strengths:**

1) Applicability to Wearable Tech: Single-channel EEG setups have potential in practical applications, and the study aims to address this need without the need to go bald you can classify Major Depressive Disorder.
2) Clear Writing: The methodology and CNN architecture design are explained concisely

**Weaknesses:**

The study’s impact is limited by its single dataset evaluation and absence of critical classification metrics :
1) Omission of Key Metrics: Essential performance metrics (F1-score, precision, recall and AUC-ROC curves) are missing, reducing evaluation comprehensiveness.
2) Underperformance: The model’s 88% accuracy is lower than Bachmann et al. (2018), which achieved 92% with a classical machine learning approach on channel Pz. And what are further advamatges of
3) Single Dataset Limitation: Evaluating on only one dataset limits generalizability, especially when other datasets like MODMA are available.
4) Lack of Comparison: Similar single-channel or CNN-based approaches, such as DOI:10.1142/S0219622019500342 and https://doi.org/10.1177/1550059420916634, could serve as useful baselines but are not compared.

**Questions:**

1) How does the model perform on a more diverse and comprehensive dataset like MODMA?
2) Could the authors clarify the omission of F1-score, precision, and recall? Including these metrics would enhance diagnostic reliability insights
3) How does this approach compare with studies like Bachmann et al. (2018) on channel Pz and other single-channel EEG studies, such as DOI:10.1142/S0219622019500342 and https://doi.org/10.1177/1550059420916634 on other important classification metrics.

---

### Official Review · Reviewer_F2rC · 2024-11-01

**Soundness:** 3
**Presentation:** 3
**Contribution:** 1
**Rating:** 3
**Confidence:** 1

**Summary:**

The authors suggest an optimized single EEG channel selection and a simple convolutional deep learning model to detect Major Depressive Disorder. The authors plan to test the detection model with a single dataset input and clearly intend it to succeed in orders of magnitude involving less number of channels of EEG.

**Strengths:**

Simplicity: The authors put forward a simple model, whereby only one channel is used to perform the EEG recordings. This may decrease the complexity as well as the cost related to the acquisition of such data in clinical or practical settings.

Focused Application: The concentrated nature of this work on the channels minimalism and application on MDD with mentioning of literature on how MA design becomes an issue while assessing patients makes sense.

**Weaknesses:**

Lack of novelty: There are no original components both in methodology and application. The employment of a single EEG channel does not come as exciting or new, and neither is there any evidence of improvements or breakthroughs in model structure or methodology that would enhance the discipline.

No comparative analysis: The study does not provide any analyses on other EEG-based MDD detection systems that are even more advanced in deep learning or more fundamental based on machine learning algorithms. Such a comparative analysis would enable positioning of the proposed MDD approach within the competitive scientific market and its merits or demerits would be illustrated.

Limited dataset: The finding cannot be said to be fully conclusive since only one dataset was employed. Testing the model against several datasets would add weight to the evidence of the model application in different settings and sociocultural characteristics of the population.

Insufficient performance gains: The paper does not exhibit any significant progresses relative to the three computed parameters such as the model accuracy, computation time, and clinical importance. In the absence of these parameters, the proposed approach does not provide credible case for progress in the field or major contributions in the area of MDD detection.

**Questions:**

Comparison with Machine Learning Methods: Have you attempted any comparative analysis of your deep learning approach with traditional machine learning methods such as SVM or Random Forest applied on the same dataset? Such analysis would define if the deep learning model is actually better suited for this task than model architectures that are less complex.

Dataset and Generalization: As you employed only one dataset, how sure are you that this particular approach would transfer to other EEG datasets or other patients? Would such an effort lead you to incorporate other datasets or cross-dataset tests?

Clinical Relevance and Practicality: In the future, how do you intend the single-channel model to have clinical relevance? Or perhaps you could explain whether you conducted any studies or experiments that would in some way prove its practical effectiveness, especially in diagnostic centers, in terms of performance or efficiency gains?

---

### Meta-Review · Area_Chair_VzEd · 2024-12-19

**Metareview:**

This study explores using single-channel EEG data and deep learning for early diagnosis of major depressive disorder. The paper tackles a very interesting and important area. However, it falls significantly short with key weaknesses including the simplicity of the method and lack of novelty (using a very simple well-known method), substantially limited experiments, the need for major presentation improvements, lack of sufficient comparisons with other works, and missing key recent literature.

**Additional Comments On Reviewer Discussion:**

The paper received 1, 1, 3, 3. No rebuttal was provided.

---

### Decision · Program_Chairs · 2025-01-22

Reject